

# PresB-Net: parametric binarized neural network with learnable activations and shuffled grouped convolution

Jungwoo Shin and HyunJin Kim

School of Electronics and Electrical Engineering, Dankook University, Yongin, South Korea

## ABSTRACT

In this study, we present a novel performance-enhancing binarized neural network model called PresB-Net: Parametric Binarized Neural Network. A binarized neural network (BNN) model can achieve fast output computation with low hardware costs by using binarized weights and features. However, performance degradation is the most critical problem in BNN models. Our PresB-Net combines several state-of-the-art BNN structures including the learnable activation with additional trainable parameters and shuffled grouped convolution. Notably, we propose a new normalization approach, which reduces the imbalance between the shuffled groups occurring in shuffled grouped convolutions. Besides, the proposed normalization approach helps gradient convergence so that the unstableness of the learning can be amortized when applying the learnable activation. Our novel BNN model enhances the classification performance compared with other existing BNN models. Notably, the proposed PresB-Net-18 achieves 73.84% Top-1 inference accuracy for the CIFAR-100 dataset, outperforming other existing counterparts.

## INTRODUCTION

Developments of neural networks lead us to provide solutions in many areas. The image classification has been advanced with the advent of convolutional neural networks (CNNs) such as LeNet-5 (*LeCun et al., 1998*), AlexNet (*Krizhevsky, Sutskever & Hinton, 2012*), VggNet (*Simonyan & Zisserman, 2014*), GoogLeNet (*Szegedy et al., 2015*), ResNet (*He et al., 2016*), DenseNet (*Huang et al., 2017*), *etc*. Their remarkable successes have increased demands for implementing CNNs on lightweight devices with limited resources (*Phan et al., 2020*; *Sandler et al., 2018*). However, CNNs require large computational and storage resources, which has been a major obstacle to lightweight implementations. A CNN model in *Courbariaux et al. (2016)* binarizes its weights. BNN models (*Courbariaux et al., 2016*; *Rastegari et al., 2016*) approximate real-value computations *via* binarized quantizations of both activations and weights, thus significantly reducing computational and storage resource usages. The binarized filter weights and activations in BNN models allow convolutions to be performed *via* bitwise XNOR and bit-counting operations. Therefore,

Corresponding author
HyunJin Kim,
hyunjin2.kim@gmail.com

BNN models can achieve tremendous resource savings compared with real-valued CNN models. Although there are significant benefits in terms of hardware costs and operating speed by using BNN models, their low classification accuracies are the most critical issue when applying the BNN models. Although memory usage and computational overhead are dramatically reduced in BNNs, classification performance can be significantly degraded. For example, the binarized ResNet-18 (*Rastegari et al., 2016*) has classification results lower than that of real-valued ResNet-18 by 18.1%. Since then, various BNN models (*Rastegari et al., 2016*; *He et al., 2018*; *Lin, Zhao & Pan, 2017*; *Liu et al., 2018*; *Bulat & Tzimiropoulos, 2019*; *Zhuang et al., 2019*; *Chakraborty et al., 2019*; *Bethge et al., 2019*; *Phan et al., 2020*; *Bethge et al., 2020*; *Liu et al., 2020*) have been proposed to overcome performance degradation of BNN models. Recently, ReActNet (*Liu et al., 2020*), one of the latest existing studies, has just lower than 3.0% of ResNet-18. Notably, Bi-Real-Net (*Liu et al., 2018*) shows that the shortcut connection per each binarized convolutional layer and weight initialization can help increase the classification accuracies of BNN models. Recently, ReActNet (*Liu et al., 2020*) dramatically increases the classification accuracies of BNN models. Besides, AresB-Net (*Kim, 2021*) adopts both shortcut addition and concatenation to increase classification accuracy. However, there are still significant performance degradations compared to real-valued counterparts.

For achieving higher classification accuracy, this paper proposes a novel BNN model called PresB-Net using learnable activation in the shuffled grouped convolutions. Besides, this proposed BNN model adopts a new normalization approach that applies batch normalization globally after applying layer normalization for each group. The proposed normalization approach can show better learning results by considering the association of groups and eliminating distribution imbalances that can arise between groups within a channel shuffle and grouped convolution. When applying a learnable activation that consists of the ReLU (Rectified Linear Unit) activation function with trainable slope and scaling parameters, it is difficult to provide convergence of gradients in the learning step when more parameters are added. However, our new normalization approach helps the gradient convergence. Our classification performance is improved by using additional reverse parametric ReLUs (RPReLU) in *Liu et al. (2020)*. In our experiments, the PresB-Net model based on the ResNet-18 (*He et al., 2016*) results in the classification accuracy of 73.84%, which achieves a performance improvement of 3.04% compared with the ReActNet model (*Liu et al., 2020*).

## RELATED WORKS
### Binarized neural networks
As the complexity of neural networks becomes higher, memory requirements and computing costs increase, posing a considerable burden on power-hungry systems. Highly quantized CNN-based models significantly reduce the required storage size and hardware costs. BNN models can achieve lightweight CNNs by quantizing either weights (*Courbariaux et al., 2016*), activations (*Hubara et al., 2016*), or both (*Rastegari et al., 2016*) into into $\{+1, -1\}$ in inference step. Thus, the BNN models replace floating-point operations with

binarized ones, approximating a bundle of floating-point multiply-accumulate operations into bitwise XNOR and bit counting operations. The XNOR-Net model in *Rastegari et al. (2016)* achieves $\approx 32\times$ storage saving and $\approx 58\times$ computation speedup, thus proving that the BNNs can be a realistic quantized CNN approach for power-hungry mobile systems. Although BNN models can significantly reduce hardware costs and power consumption, highly degraded classification accuracies limit their applications.

Several approaches presented training methods and neural network structures for BNN models. Firstly, a training method for BNNs was introduced in *Courbariaux, Bengio & David (2015)*. The XNOR-Net (*Rastegari et al., 2016*) presented a BNN basic block for generating binarized activations and scaled convolution outputs. The basic block in the binarized CNN models showed the validity of this new BNN structure empirically. However, their degraded accuracies seemed to be substantial compared with those from real-valued models.

Until now, many works have focused on enhancing the performance of BNN models. Several works proposed novel BNN structures to solve this problem of BNN models. Notably, many state-of-the-art BNN models were based on residual networks using so-called shortcuts. The binarized ResNet using the XNOR-Net scheme (*Rastegari et al., 2016*) contained shortcuts per two binarized convolutional layers, like the original ResNet model (*He et al., 2016*). In *Liu et al. (2018)*, unlike the original ResNet model, each binarized convolutional layer has a shortcut. The shortcuts increased resolutions in each binarized convolutional layer, amortizing the error from binarizations in *Liu et al. (2018)*. Several BNN models were motivated by other successful CNN models. The binarized depth-wise separable convolution in *He et al. (2018)* and *Phan et al. (2020)* adopted grouped convolutions and increased the number of channels. In *Bethge et al. (2019)* and *Bethge et al. (2020)*, shortcuts were concatenated to expand channels motivated from the DenseNet (*Huang et al., 2017*) model. In AresB-Net (*Kim, 2021*), the grouped convolutions and shortcut concatenations were performed with shuffled channels. In *Liu et al. (2020)* and *Wang et al. (2020a)*, the binary activation layers with a learnable shift were proposed. After reviewing these existing works, we conclude that specific structures and layers for BNN models help increase classification accuracies.

Several works considered training scheme and parameter optimization for BNN models in *Alizadeh et al. (2018)*, *Zhu, Dong & Su (2019)*, *Wang et al. (2019)*, *Hubara et al. (2017)*, *Ghasemzadeh, Samragh & Koushanfar (2018)*, *Gu et al. (2019a)*, *Helwegen et al. (2019)*, *Ding et al. (2019)*, *Martinez et al. (2020)*, *Kim et al. (2021)* and *Chen et al. (2021)*. Although these specific methods can increase the performance of BNN models, we focus on a new BNN structure and do not consider specific training optimization schemes in this paper.

## Shortcut and shuffled grouped convolution

In residual neural networks, convolutional layers are piled up, skip connections called *shortcuts* make input features of a layer merged with the outputs of a convolutional layer. The shortcut introduced in *He et al. (2016)* can bypass the block layer, so that it allows CNNs to achieve fast training models. Besides, the shortcut provides ensemble-like effects and reduces the vanishing gradient effect. This method in *He et al. (2016)* shows the

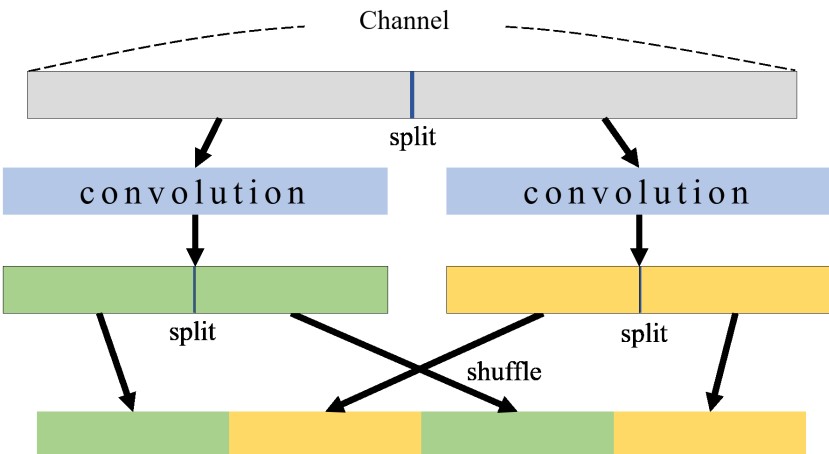

**Figure 1   Group convolution and shuffle.**

skip connection by adding the skipped features to the convolutional output, so that the number of channels is not increased. On the other hand, DenseNet (*Huang et al., 2017*) concatenates skip connection to the convolutional output, thus increasing the number of channels. Recently, AresB-Net (*Kim, 2021*) uses a mix of two skip connection methods, expanding the number of channels without additional computational costs.

In a CNN model, channels mean feature maps that contain image information, so that channels allow us to provide desired information from the model. As the number of channels increases, more image information can be available in neural networks. BNNs have a weak point in that weights and features are quantized into 0 or 1. ShuffleNet (*Zhang et al., 2018*) uses grouped convolution and shuffles channels along with expanding channels. The grouped convolution and shuffle process for two groups is illustrated in Fig. 1. In grouped convolutions, channels are divided for each group to perform convolutions in each group separately, which significantly reduces computational costs. When expanding channels, grouped convolutions can provide more features without increasing computational costs. However, if only a grouped convolution is applied, there is no interaction between groups which weakens the representation of the information. The channel shuffle (*Zhang et al., 2018*) can address this problem, mixing channels between groups. In the channel shuffle, channels within each group are divided into subgroups, and subgroups from different groups are gathered into a new group. A channel shuffle can mix channels between groups so that features of all groups can be associated with each other in grouped convolutions. This shuffled grouped convolutional network is called a shuffle network.

## Learnable activation

ReActNet (*Liu et al., 2020*) presents a new approach called the reverse parametric rectified linear unit activation function (RPReLU). RPReLU uses PReLU activation and learnable biases together.

PReLU (*He et al., 2015*) activation function learns a slope parameter for negative values and updates the parameter during training. The learnable parametric slope enables us

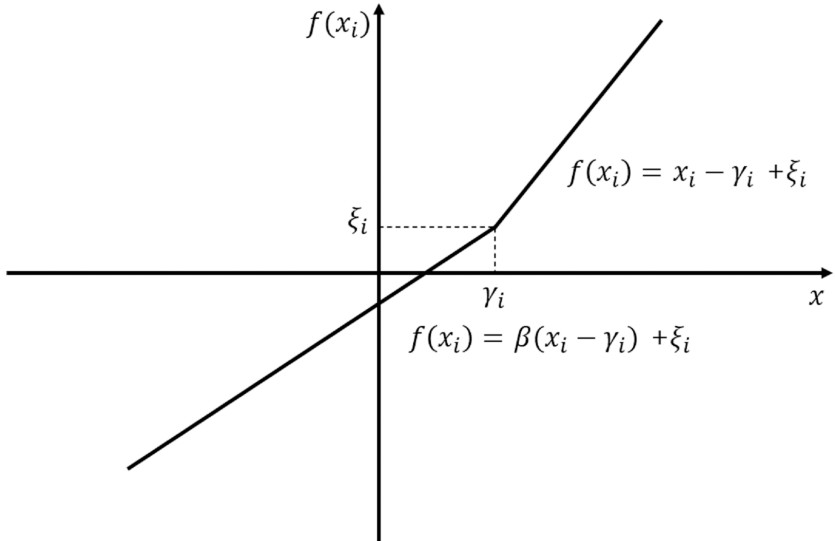

**Figure 2** **Graphical representation of RPReLU (*Liu et al., 2020*).**

to proceed with more suitable activations. RPReLU adds learnable biases to the PReLU activation. The learnable biases are located both before and after the PReLU activation. Overall, the binarized convolutions in BNNs could shift their output distribution compared with real-value counterparts because features are quantized into $-1, 1$. It is known that RPReLU amortizes the imbalanced distribution using additional learnable biases, thus providing better classification performance.

RPReLU is formulated in Eq. (1), where $i, x_i$ denote the channel index and a feature of $i$th channel, respectively. Besides, terms $\gamma_i, \zeta_i$ are learnable biases to make distribution more balance. Eq. (1) is illustrated in Fig. 2. Term $\beta$ denotes the learnable parametric slope to control the negative part slope originated from PReLU.

$$f(x_i) = \begin{cases} x_i - \gamma_i + \zeta_i & if \; x_i > \gamma_i \\ \beta(x_i - \gamma_i) + \zeta_i & if \; x_i \leq \gamma_i \end{cases}. \tag{1}$$

### Batch normalization

Batch normalization (*Ioffe & Szegedy, 2015*) can stabilize the learning process and accelerate the learning speed. A training set is divided into several subsets called mini-batches in batch normalization. During training, the reason for the learning instability is due to the *rough optimization landscape* (*Santurkar et al., 2018*). To address this, batch normalization calculate the mean and variance from the mini-batch and normalizes features using them for each channel. This normalization process makes the optimization landscape smoother, achieving a more stable learning step (*Santurkar et al., 2018*).

The batch normalization process to normalize in a mini-batch is formulated in Eq. (2). Terms $\gamma$ and $\beta$ denote trainable parameters. Term $M$ denote the size of mini-batch $B$, where $\mu^B$ and $\sigma^B$ denote the mean and variance of the mini-batch $B$. Features $x_i$ are normalized based on $\mu^B$ and $\sigma^B$. In Eq. (2), term $\hat{x}_i$ denotes the normalized values by

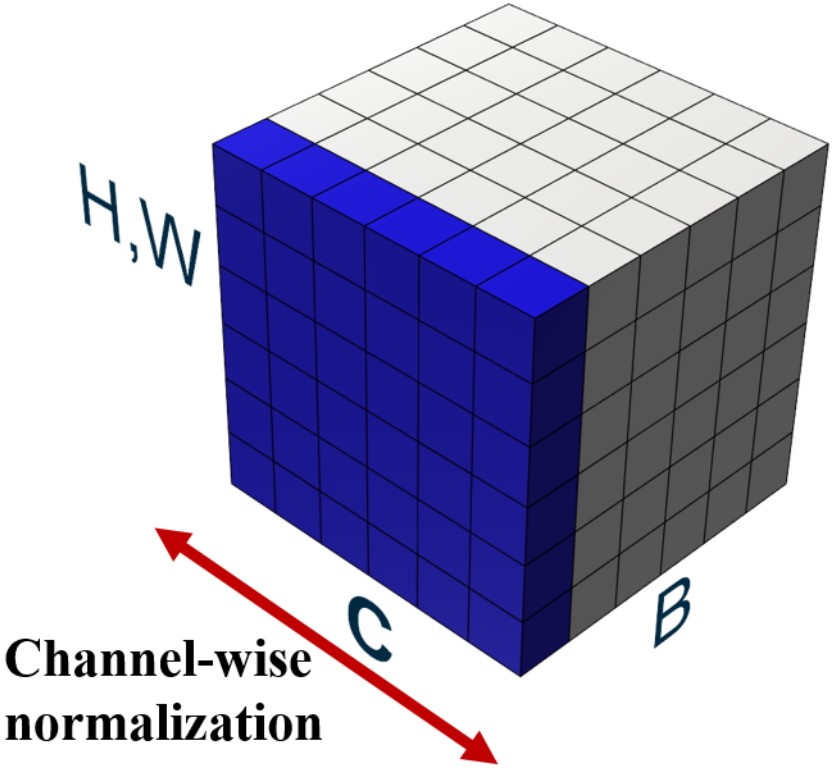

**Figure 3** **Illustration of adopted layer normalization.**

using $\mu^B$, $\sigma^B$. Term *epsilon* means a small value (*e.g.*, 1e−5) that is used to to prevent the divide-by-zero error. After scaling and biasing $\hat{x}_i$ with $\gamma$ and $\beta$, $BN_{\gamma,\beta}(x_i)$ is the output of the batch normalization process.

$$B = \{x_1, x_2 \cdots x_M\}$$

$$\mu^B = \frac{1}{M}\sum_{i=1}^{M} x_i \qquad \sigma^B = \sqrt{\frac{1}{M}\sum_{i=1}^{M}(x_i - \mu^B)^2}$$

$$\hat{x}_i = \frac{x_i - \mu^B}{\sqrt{(\sigma^B)^2 + \varepsilon}} \qquad BN_{\gamma,\beta}(x_i) \equiv \gamma\hat{x}_i + \beta. \tag{2}$$

*Layer normalization*

Layer normalization (*Ba, Kiros & Hinton, 2016*), unlike batch normalization, does not depend on batches and normalizes features across channels. In the layer normalization process, both mean and variance are computed across all channels and a spatial feature dimension is shown in Fig. 3. In Fig. 3, feature maps are illustrated with $H, W$ as the spatial axes, $B$ as the batch axis, and $C$ as the channel axis, respectively. A batch consists of multiple images, which are used as input at a time during training. In layer normalization,

the features of these entire channels are normalized for a single image so that the normalized result for a single image does not depend on features of other images in a batch.

The formulation of the layer normalization process is shown in Eq. (3). Term $H$ denote the number of features along with the feature direction. Similar to batch normalization in Eq. (2), $\mu^l$ and $\sigma^l$ denote the mean and variance of a layer. After calculating $\mu^l$ and $\sigma^l$, features are normalized using $\mu^l$ and $\sigma^l$. Term $\hat{a}_i$ denotes the normalized value and adding bias in $\hat{a}_i$ is the output of the layer normalization process. After scaling and biasing $\hat{a}_i$ with $\gamma$ and $\beta$, the layer normalization process outputs $LN_{\gamma,\beta}(x_i)$.

$$\mu^l = \frac{1}{H}\sum_{i=1}^{H}a_i^l \qquad \sigma^l = \sqrt{\frac{1}{H}\sum_{i=1}^{H}(a_i^l - \mu^l)^2}$$

$$\hat{a}_i = \frac{a_i - \mu^l}{\sqrt{(\sigma^l)^2 + \varepsilon}} \qquad LN_{\gamma,\beta}(a_i) \equiv \gamma\,\hat{a}_i + \beta \tag{3}$$

## PROPOSED BNN MODEL

The proposed BNN model applies a new normalization approach by using both batch and layer normalization. For high classification accuracy, we take the state-of-the-art BNN structures described in *Kim (2021)* and *Liu et al. (2020)*. The proposed BNN model expands channels by using grouped convolution and uses a shuffle network to mix channels of different groups. Besides, the proposed model adopts the learnable bias and activation. In this section, the proposed model structure and its adopted blocks are described in detail. Besides, the normalization and training methods are explained.

### Binarized convolution

In our BNN model, the binarized convolution can handle operations by binarized weights and features. The binarized convolution performs its filtering using bitwise XNOR and bit-counting operation, rather than real-valued multiplications and accumulations. Binary activation can be performed using *sign* function. In Eq. (4), input $x$ quantifies the values to 0 and 1 based on the negative positive number.

$$x \in \{I\}, \qquad Sign(x) = \begin{cases} 1 & x \geq 0 \\ -1 & x < 0 \end{cases}, \tag{4}$$

Binarized convolutions can be shown in Eq. (5).

$$Sign(I) \in \{1,-1\}^{C_i \times w \times h} \qquad Sign(W) \in \{1,-1\}^{C_i \times C_o \times w \times h}$$

$$BConv(I,W) = Sign(I) \circledast Sign(W) \odot \alpha. \tag{5}$$

Superscripts $C_i \times w \times h$ and $C_i \times C_o \times w \times h$ are the number of vector dimensions. Term $\alpha$ denotes the scaling factor for weights, which can be $\frac{1}{C_i \times w \times h}$ in *Rastegari et al. (2016)*.

Symbols ⊛ and ⊙ denote the binarized convolution using bitwise XNOR & bit-counting operations, and element-wise scalar multiplication, respectively. In other words, each weight is scaled by being multiplied with $\alpha$. After binarizing weights, the multiplication with the binarized activations is approximated using the bitwise XNOR operation. Terms $C_i$ and $C_o$ denote the numbers of input channels and output channels. Terms $w$ and $h$ mean the width and height of a feature map. For an image classification, $Sign(I)$ outputs $(C_i \times w \times h)$-dimensional binary value of $-1$ or $1$. Term $BConv(I, W)$ denotes binarized convolution with binarized inputs and binarized weights.

Instead of conventional convolution, we use grouped convolution. Eq. (6) formulates how the grouped binarized convolution works when the number of groups is two, where the term $GBConv$ denotes the grouped binarized convolution. After input $I$ is quantized into $\{1, -1\}$, it is divided into two groups $I_1$ and $I_2$ of the channels. After the grouping, the binarized convolution is performed with each group. Each binarized convolution adopt weight $W_1$ and $W_2$, respectively, which is also quantified to $1, -1$. The two binarized convolution results, $BConv(I_1, W_1)$ and $BConv(I_2, W_2)$, are concatenated for achieving the final grouped binarized convolution result denoted as $GBConv(I, W)$. Symbol $\oplus$ indicates the concatenation for producing the final result.

$$Sign(I_1) \in \{1, -1\}^{\frac{C_i}{2} \times w \times h}, \qquad Sign(I_2) \in \{1, -1\}^{\frac{C_i}{2} \times w \times h}$$
$$Sign(W_1) \in \{1, -1\}^{\frac{C_i}{2} \frac{C_o}{2} wh}, \qquad Sign(W_2) \in \{1, -1\}^{\frac{C_i}{2} \frac{C_o}{2} wh}$$
$$GBConv(I, W) = BConv(I_1, W_1) \oplus BConv(I_2, W_2) \tag{6}$$

In the grouped binarized convolution, the number of parameters is $\frac{1}{2} \times C_i \times C_o \times w \times h$ while the conventional convolution has $C_i \times C_o \times w \times h$ parameters. Therefore, with only half the parameters, the grouped binarized convolution can reduce the required number of parameters, which allows the computation of doubled input channels without additional computational costs.

## Proposed normalization approach

Activation is carried out after performing the grouped binarized convolution. The grouped convolution makes the activation output have the imbalanced distribution and produces rough optimization landscape (*Santurkar et al., 2018*). The imbalanced distribution and rough optimization landscape make gradients hard to converge in the learning step. In a typical CNN, normalization prevents this distribution imbalance and difficulty of gradient convergence.

Our proposed new normalization technique overcomes the aforementioned difficulties by making the optimization landscape smooth in the learning process and considering the relationship between channels in grouped convolution. As described in our binarized convolution, the output of the convolutional layers are divided into multiple groups. Firstly, our proposed approach performs layer normalization in a channel-wise manner. Layer normalization proceeds the normalization of values over the entire channels, adjusting the outputs of all group convolutions to the same scale. After finishing layer normalization, batch normalization is performed on the values from the layer normalization layer. This

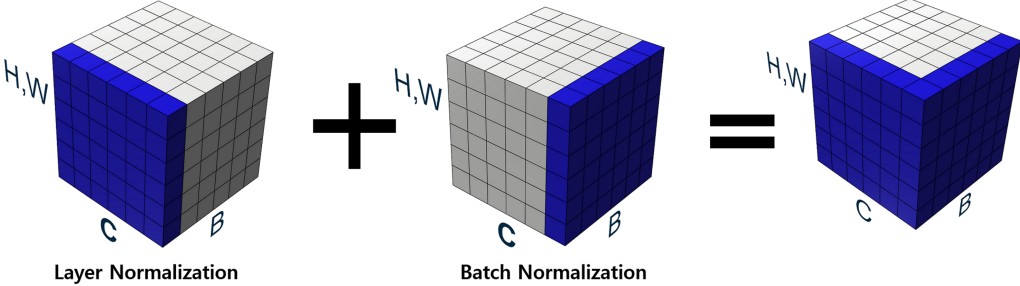

**Figure 4 Proposed normalization.**

process is illustrated in Fig. 4, which illustrates $H, W$ as the spatial axes, $B$ as the batch axis, and $C$ as the channel axis, respectively. The blue-colored part represents features to be normalized.

All channels are considered in channel-wise layer normalization so that the layer normalization is not restricted in the channels of a group. The layer normalization calculates both mean and variance, which are applied to each channel. Relationships between one group and another group can also be considered. Besides, because layer normalization considers the features of other groups, features on each group are normalized by deviation of the features from all groups. Therefore, even though significant imbalances exist between features of different groups, the imbalance could be smoothed, which makes it possible to alleviate abrupt changes in features between groups. If the imbalance between features of groups is too significant, the imbalance should be reduced to maintain the difference as shown in Fig. 5. If the imbalance between groups is too small, it makes groups balanced on the same scale while maintaining the difference, as shown in Fig. 6. This method makes groups have the same scaling factor for balancing to eliminate imbalance, no matter what imbalance exists after grouped convolution. Next to layer normalization, additional activation and batch normalization are placed, which is illustrated in Figs. 7 and 8 of *Basic and expand blocks* subsection.

Therefore, the grouped binarized convolution results are normalized in both channel-wise and batch-wise manner. The proposed normalization approach allows the normalized convolutional results in both channel-wise and batch-wise manners, which can consider data of other groups to normalize and relieve imbalance between groups.

In our proposed normalization, the biased PReLU for a nonlinear activation with learnable parameters is adopted. The biased RPeLU contains a learnable bias for each channel and then performs the conventional PReLU (*He et al., 2015*). PReLU has been used in various existing models such as ReActNet (*Liu et al., 2020*) and PReLU-Net (*He et al., 2015*). Considering our evaluation results, PReLU increases classification performance by adding parametric slope values in ReLU. Unlike RPReLU (*Liu et al., 2020*), term $\zeta_i$ of Eq. (1) is not required in the biased PReLU. Eq. (7) formulates the proposed normalization. In Eq. (7), $i_{hw}$ denotes an input vector of channels when its position is $(h, w)$ on the height and width axes of a feature map. Terms $g$ denotes a group, and term $\gamma_g$ denotes the learnable bias for the group $g$. Term $\beta$ denotes the learnable parametric slope of the biased

**Peer**J Computer Science

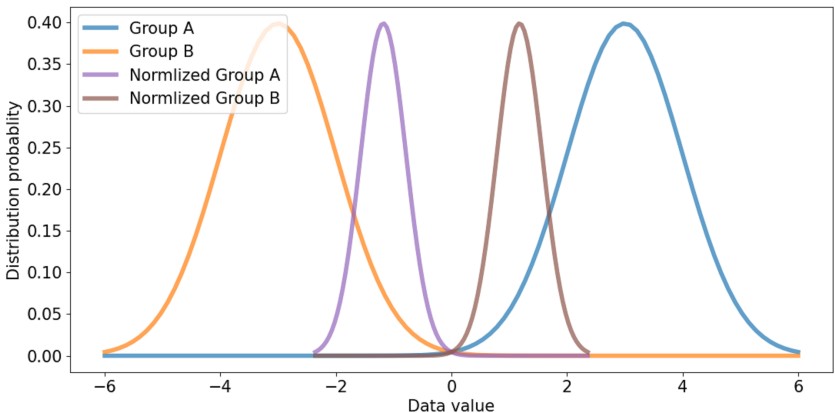

**Figure 5** Large distribution gap group and scaled balanced group.

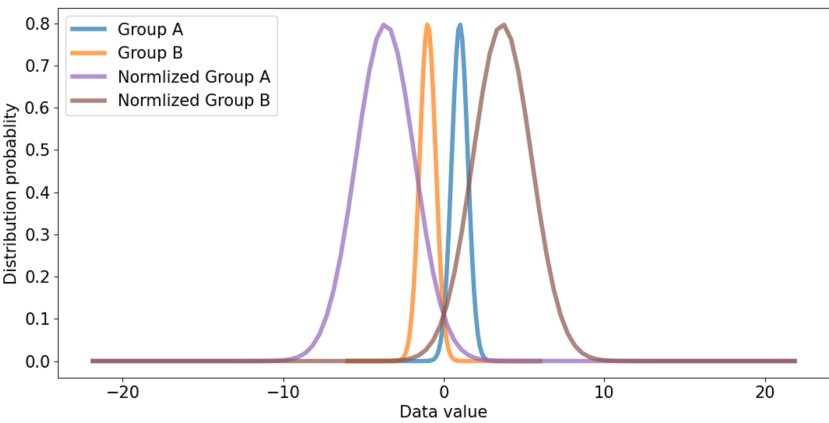

**Figure 6** Small distribution gap and scaled balanced group.

PReLU. these values are calculated for each group. Applying bias $\gamma_g$ and PReLU with slope $\beta$, $f(x_i)$ is the result of biased PReLU. These results conduct layer normalization. It solves the imbalance between groups arising from shuffled grouped convolution by considering the other group data and allows the results of each group have the same balanced output, thereby increasing performance by the balanced group and increasing the convergence.

$$f(x_i) = \begin{cases} x_i - \gamma_g & if \ x_i > \gamma_g \\ \beta(x_i - \gamma_g) & if \ x_i \le \gamma_g \end{cases}, x_i \in X_i. \tag{7}$$

## Basic and expand blocks

As shown in *Rastegari et al. (2016)*, *Liu et al. (2018)*, *Liu et al. (2020)* and *Kim (2021)*, BNNs with shortcuts can provide good performance. The proposed basic block can be suitable for the stacked structure such as ResNet (*He et al., 2016*). In the following description, the proposed blocks are applied to the stacked structure. In our model, the downsampling

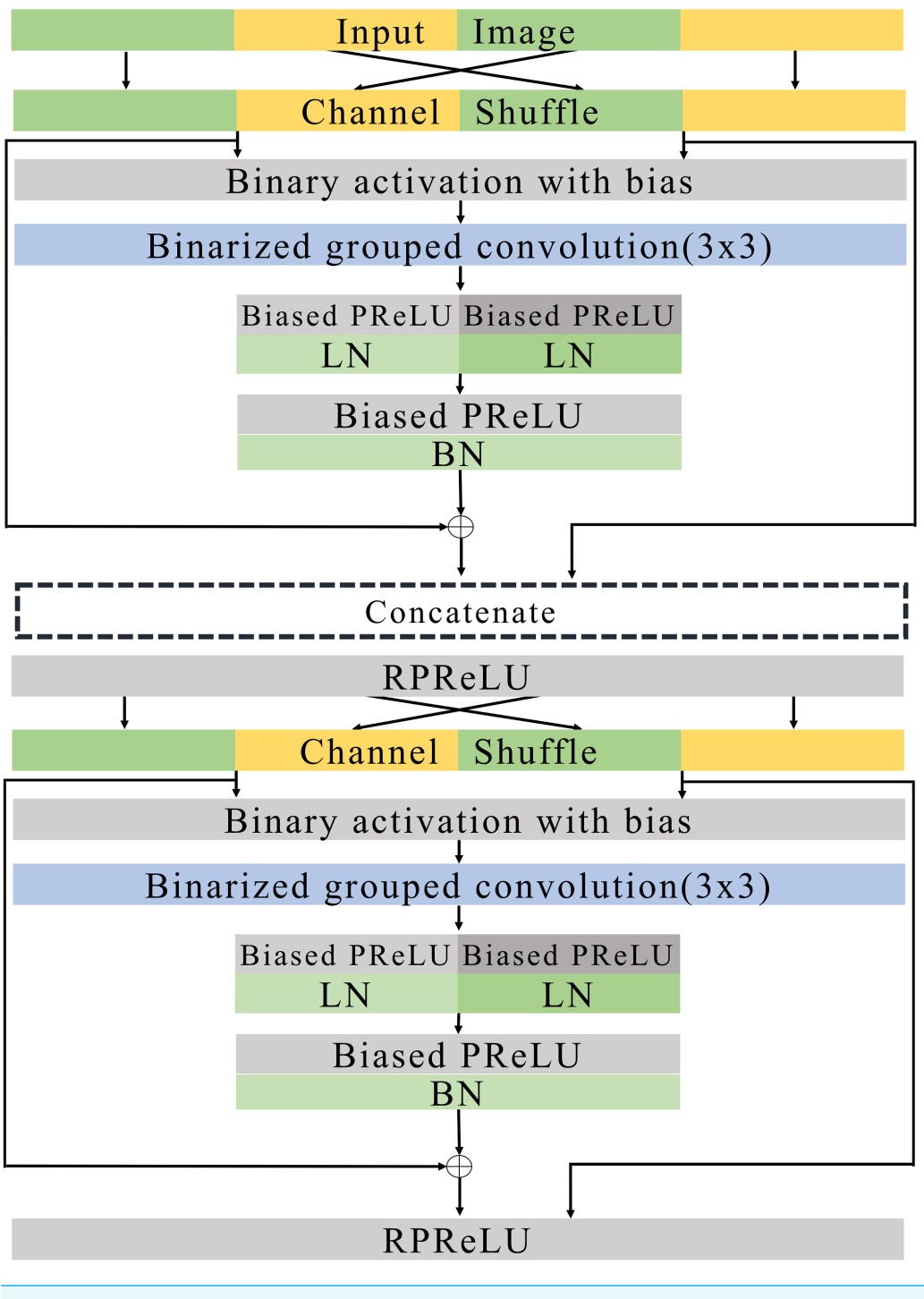

**Figure 7   Basic blocks.**

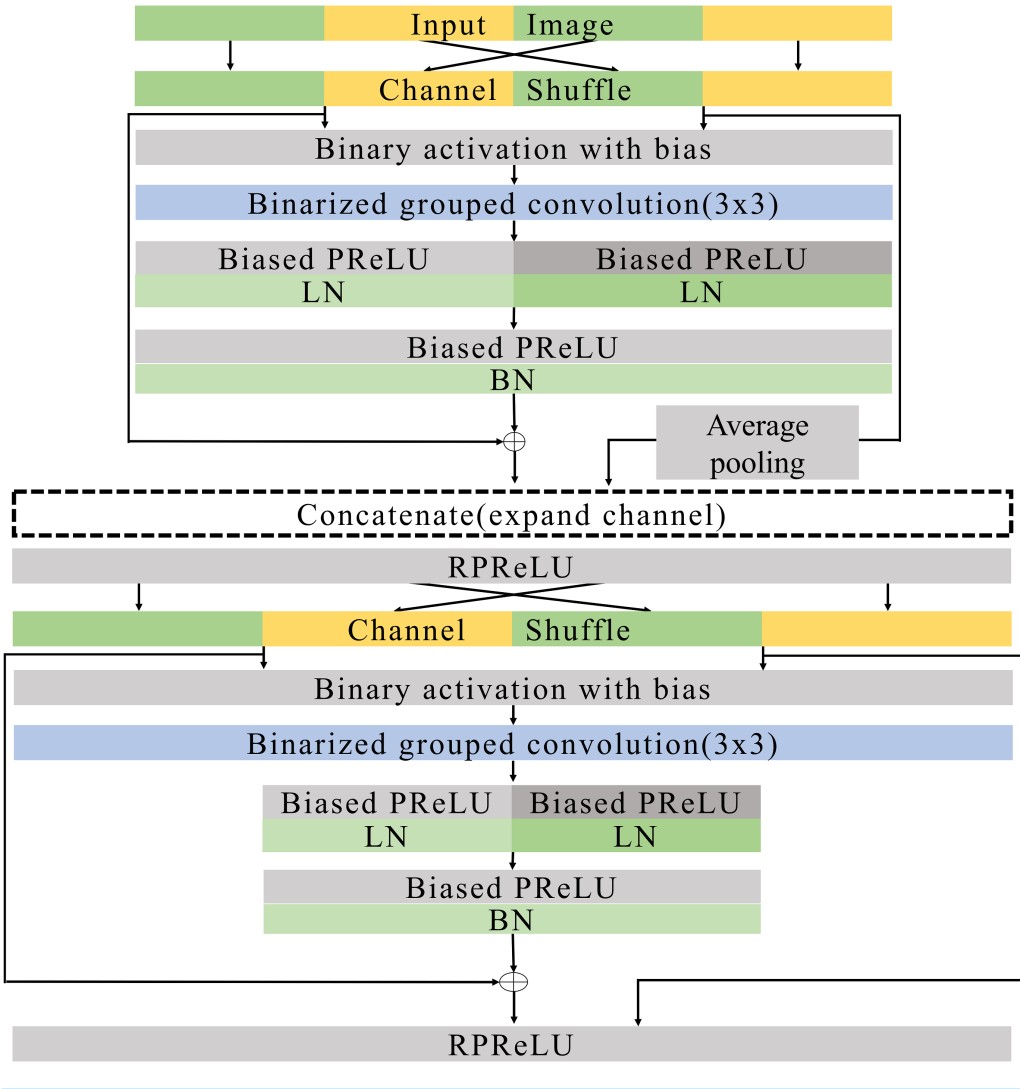

**Figure 8  Expand blocks.**

reduces the width and height of an input feature map with the factor of stride as 2. At the same time, the number of channels is doubled.

Figures 7 and 8 illustrate the basic and expand blocks. The blocks contain two binarized grouped convolutional layers. After proceeding grouped convolution layer, the channel is shuffled. This grouped convolution and channel shuffle are used to increase the channel with the same amount of computation in ShuffleNet *Zhang et al. (2018)* and AresB-Net in *Kim (2021)*. The channel expansion and binarized grouped convolution adopt the structure in *Kim (2021)*. Similarly, we adopt binarized grouped convolution rather than grouped convolution for binarized operation. The input and output channels of a basic block have the same feature map size. Besides, the numbers of input and output channels are the same. On the other hand, the number of output channels is doubled over that of input channels in an expand block. Besides, the width and height of an input feature map are

reduced by half. In both blocks, after shuffling channels, the binary activation layer with the learnable bias for each channel is located. The learnable bias for each channel is added to input features. Then, the values are binarized based on Eq. (4). Each block performs its first binarized grouped convolutions with $3 \times 3$ kernels for two groups. The number of output channels in the first binarized grouped convolution depends on the stride. If the stride for the basic block is one, there is no downsampling in the first convolution. In the expand block, the downsampling is performed in the first binarized grouped convolution with the factor of stride as 2.

Terms $C_i$ and $C_o$ are denoted as the numbers of input and output channels for a block. Let $g_1$ and $g_2$ denote the group index for the grouped convolution, $C_i(g_1)$ and $C_i(g_2)$ can be $\frac{1}{2} \times C_i$, respectively, so that $C_i = C_i(g_1) + C_i(g_2)$. In the basic block, $C_o(g_1) = C_o(g_2) = \frac{1}{4} \times C_i$. Therefore, the binarized grouped convolution has $\frac{1}{2} \times C_i$ output channels, which is illustrated in Fig. 7. On the other hand, in the expand block, $C_o(g_1) = C_o(g_2) = \frac{1}{4} \times C_i$. the binarized grouped convolution has $C_i$ output channels. Figures 7 and 8 describe the doubled number of output channels in the expand block by expressing the number of channels with the length of each layer. Note that Fig. 6 has a longer length, which means that it has more output channels compared with the basic block.

Let the costs of a $3 \times 3$ kernel and two-dimensional feature map of a channel be $cost_{k,w,h}$ in our binarized convolution. Since the computation costs of a convolution are proportional to the number of input and output channels, the costs of the binarized convolution for a group is $\frac{1}{2} \times C_i \times \frac{1}{4} \times C_i \times cost_{k,w,h}$. Therefore, the computation costs can be $\frac{1}{4} \times C_i^2 \times cost_{k,w,h}$. When channels are not expanded, the conventional binarized convolutional layer (*Rastegari et al., 2016*; *Liu et al., 2018*) based on ResNet (*He et al., 2016*) can have $\frac{1}{2} \times C_i$ input channels and $\frac{1}{2} \times C_i$ output channels in their basic block. Therefore, the computation costs of the conventional binarized convolution can be also $\frac{1}{4} \times C_i^2 \times cost_{k,w,h}$. Therefore, although input channels are expanded, the computation costs do not increase by using grouped convolution in our blocks.

The proposed normalization approach is applied so that the layer and batch normalizations are performed after applying the biased PReLU, respectively. Then, two different types of shortcuts are concatenated to expand channels. Like *Kim (2021)*, the shortcut from $\frac{1}{2} \times C_i$ input channels are added to the output of batch normalization and concatenated. Besides, $\frac{1}{2} \times C_i$ input channels after the channel shuffle are concatenated so that $C_i$ channels are obtained from the channel concatenation. In the expand block, the average pooling is used for shrinking the size of each feature map. The $C_i$ channels from the average pooing are concatenated so that $2 \times C_i$ channels are prepared for the next step. Then, RPReLU (*Liu et al., 2020*) is placed, which further increases performance by adding parameters in Eq. (1).

A channel shuffle is performed on the output of this RPReLU. Then, the binary activation, binarized grouped convolution, and the proposed normalization with additional parameters are performed in order. After concatenating shortcuts and applying RPReLU, the final output is produced. While the number of output channels of the expand block is twice that of input channels in the expand block, the numbers of input and output channels in the basic block are the same.

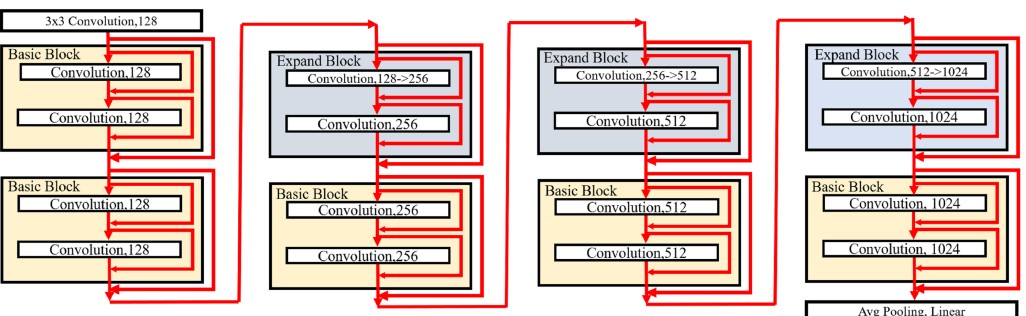

**Figure 9   Model structure of PresB-Net-18.**

Like ResNet (*He et al., 2016*), the proposed model skips two binarized grouped convolutions and add input features to the output of the final RPReLU. Because the basic block does not change the size of a feature map, the shortcut with the input features is added directly. On the other hand, the size of feature map changes in the expand block. So the shortcut contains the average pooling, $1 \times 1$ binarized convolution, and batch normalization layers. Generally, $1 \times 1$ convolutions are used in the downsampling of residual CNNs. While real-valued $1 \times 1$ convolutions are adopted in several existing BNNs (*Rastegari et al., 2016*; *Liu et al., 2018*), the proposed blocks use $1 \times 1$ binarized convolutions, which can reduce hardware costs significantly, but compared with existing ReActNet and AresB-Net, the additional $1 \times 1$ convolution for shortcuts can increase computational and storage costs. However, our proposed model can increase the classification accuracy compared with ReActNet and AresB-Net, which will be shown in *Experimenal Results* section.

## Model structure

The proposed blocks are used to build our PresB-Net, which has a pyramid structure by stacking basic and expand blocks. Figure 9 illustrates a model structure of PresB-Net containing stacked 8 blocks for the CIFAR-100 dataset as an example. The number in a small box indicates the number of output channels. In the first $3 \times 3$ convolutional layer, the real-valued convolution takes three channels for the red, blue, and green colors) from a target image and have 128 output channels. The model structure in Fig. 9 follows ResNet-18 (*He et al., 2016*) so that the model is called PresB-Net-18 with a suffix representing the number of convolutional layers. The number of binarized grouped convolutional layers is 16. After finishing the final basic block, the average pooling is applied. The fully connected linear layer is used to produce classification results. In total, 18 convolutional layers are placed in the example. Following Figs. 7 and 8, the red arrows inside blocks indicate the direction of features including shortcuts for each binarized grouped convolution. The red arrows outside blocks illustrate the direction of features and shortcuts that skip two binarized grouped convolutions for each block.

The downsampling is performed in the expand blocks. The model shown in Fig. 9 has 1,024 output channels after the final basic block. After performing 4 binarized grouped convolutions, the channels are expanded. PresB-Net-18 model structure can be represented

as PresB-Net $(2, 2, 2, 2)$ in detail, which means that 2 blocks have the same number of output channels. Besides, PresB-Net $(2, 2, 2, 2)$ indicates that PresB-Net-18 has 8 blocks in total. The width and height of each feature map are reduced by half in the downsampling. Various models can be built depending on the number of stacking blocks and downsamplings and datasets. For example, PresB-Net-10 and PresB-Net-34 are represented as PressB-Net $(1, 1, 1, 1)$ and PresB-Net $(3, 4, 6, 3)$, respectively.

### Training and Inference

The conventional training method in *Rastegari et al. (2016)* is adopted in the training. During the training, real-valued weights are updated by backward propagation. Then, kernel weights are binarized for the binarized convolutions and then used in the forward pass. Besides, the binarized grouped convolutions take the binarized input features, as shown in Eq. (4). The derivative of the binary activation using *sign*() function is approximated with the baseline *straight-through-estimator* in *Courbariaux et al. (2016)*. Binary weights for the binarized grouped convolutions are only maintained for the inference, which can reduce storage resource requirements.

For better classification performance, pre-trained real-valued weights are used to initialize model parameters, as known in *Lin, Zhao & Pan (2017)*, *Liu et al. (2018)* and *Liu et al. (2020)*. In this case, binary activation layers are not placed. Besides, real-valued grouped convolutions replace the binarized grouped convolutions in the target model. After pretraining, the real-valued model parameters with the best accuracy are obtained and are binarized in the initialization.

## EXPERIMENTAL RESULTS

### Experimental setups

We evaluated the proposed model and other counterparts on the CIFAR datasets (*Krizhevsky & Hinton, 2009*). The CIFAR dataset has 60,000 $32 \times 32$ color images. The training and validation adopt 50,000 and 10,000 images, respectively, where CIFAR-10 and CIFAR-100 contain 10 and 100 different classes. In the data augmentation, each image in CIFAR-10 and CIFAR-100 was padded with zeros to make a $40 \times 40$ image. Then, the random cropping was applied to obtain a $32 \times 32$ image, and we horizontally flipped the images with $p = 0.5$ probability. Augmentation is not adopted during inference.

For initialization, real-valued weights were obtained with 400 epochs. we used the Adam optimizer (*Kingma & Ba, 2014*) with the learning rate of 0.0005 and weight decay of 0.0001. The real-valued weights were used in the initialization of the binarized model. Then, 400 epochs were performed again with the binarized model, where the Adam optimizer with a learning rate of 0.0005 was adopted. Notably, we did not apply weight decay because the binarization can provide enough regularization (*Liu et al., 2018*; *Liu et al., 2020*).

We trained PresB-Net-10, PresB-Net-18, and PresB-Net-34 on the CIFAR-10 and CIFAR-100 datasets. The details of the above models are explained in *Model structure* subsection.

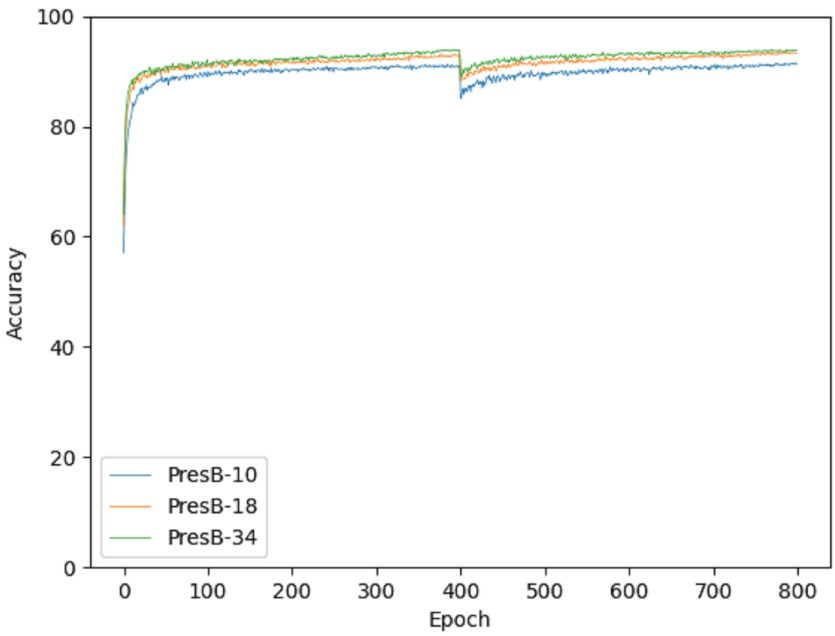

**Figure 10** **Training graph on CIFAR-10 dataset.**

## Experimental results

Accuracy graphs in the learning process are shown in Figs. 10 and 11 on the CIFAR-10 and CIFAR-100 datasets. Each graph shows the accuracy of PresB-Net-10, PresB-Net-18, and PresB-Net-34 for each learning epoch. As shown in these graphs, the number of overall training epochs was set to 800. As previously explained, the first 400 epochs initialized weights for the real-valued convolutions, and the next 400 epochs proceeded with the binarized convolutions and activations. The results of the comparison model AresB-Net, ReActNet, and proposed model PresB-Net can be seen in Table 1 when initialization was performed to real values for the CIFAR-100 dataset. The real-valued parameters were binarized from the 400-th epoch, so that accuracy decreased sharply and then increased again. At the final epoch, the accuracies were very close to those with real-valued parameters. Besides, as the number of stacked layers increased, the proposed model can achieve higher classification accuracy. It was concluded that the proposed model structures can be well trained on the described learning process. Figures 10 and 11 show the characteristics of the proposed model for obtaining better performance along with increasing number of stacked layers.

For fair comparisons, we applied the same hyperparameters of the learning process and training method to baseline AresB-Net and ReActNet. We used Adam optimizer which requires little tuning and conducted without complex and detailed parameter tuning in the learning process, so comparison results show lower accuracy than the results of the model presented in AresB-Net (*Kim, 2021*). In this training method, we initialize the model with real values. The results of real value initialization are shown in Table 1. Although the proposed PresB-Net-18 shows a −3.3% decline in performance compared

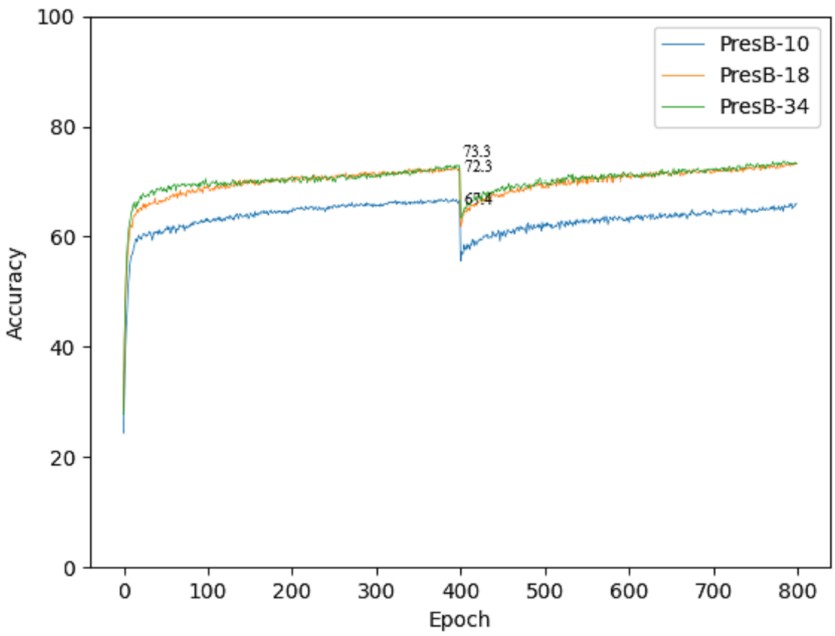

**Figure 11** Training graph on CIFAR-100 dataset.

**Table 1 Comparisons in terms of top-1 accuracies (%) on real value initialization (400 epochs).**

| Dataset | Baseline | Top-1 | Top-5 | Top-1 gap |
|---------|----------|-------|-------|-----------|
| CIFAR-100 | ResNet-18 | 75.61 | 93.05 | −3.3 |
| | AresB-Net-18 | 71.98 | 90.29 | 0.3 |
| | ReActNet-18 | 70.76 | 88.44 | 1.5 |
| CIFAR-100 | PresB-Net-10 | 67.4 | 85.9 | – |
| | PresB-Net-18 | 72.3 | 89.8 | – |
| | PresB-Net-34 | 73.3 | 89.3 | – |

to ResNet-18 created for real value operation purposes, PresB-Net-18 results show 72.3% Top-1 accuracy, which is 0.3% and 1.5% higher than 71.98% of AresB-Net and 70.76% of ReActNet, respectively.

PresB-Net shows higher accuracy for real value initialization, which is the basis for higher accuracy results for binary operations after initialization. In comparison, our proposed model shows better results in terms of Top-1 classification accuracy. These proposed model training results are shown in Table 2. In Fig. 12, PresB-Net-18 achieves 73.8% Top-1 accuracy, which is 2.8% and 3.0% higher than 71.0% of AresB-Net and 70.8% of ReActNet, respectively. Besides, PresB-Net-10 achieves 67.03% Top-1 accuracies on the CIFAR-100 dataset, which are 1.6% and 0.3% higher than 65.4% of AresB-Net-10 and 66.7% of ReActNet-10, respectively. PresB-Net-34 achieves 73.56% Top-1 accuracies on the CIFAR-100 dataset, which are 1.9% and 2.7% higher than 71.6% of AresB-Net-34 and 70.9% of ReActNet-34, respectively. Compared with AresB-Net, it is expected that biased PReLU and the new proposed normalization improved its classification

Table 2 Summary of accuracies (%) on CIFAR-10, CIFAR-100 and ImageNet datasets.

| Dataset | Model | Final Top-1 (%) | Best Top-1 (%) | Top-5 (%) |
|---|---|---|---|---|
| CIFAR-10 | PresB-Net-10 | 91.76 | 91.81 | – |
| | PresB-Net-18 | 93.36 | 93.57 | – |
| | PresB-Net-34 | 93.64 | 93.87 | – |
| CIFAR-100 | PresB-Net-10 | 67.03 | 67.03 | 85.83 |
| | PresB-Net-18 | 73.84 | 73.87 | 90.55 |
| | PresB-Net-34 | 73.56 | 73.93 | 90.39 |
| ImageNet | PresB-Net-18 | 63.03 | 63.03 | 83.70 |

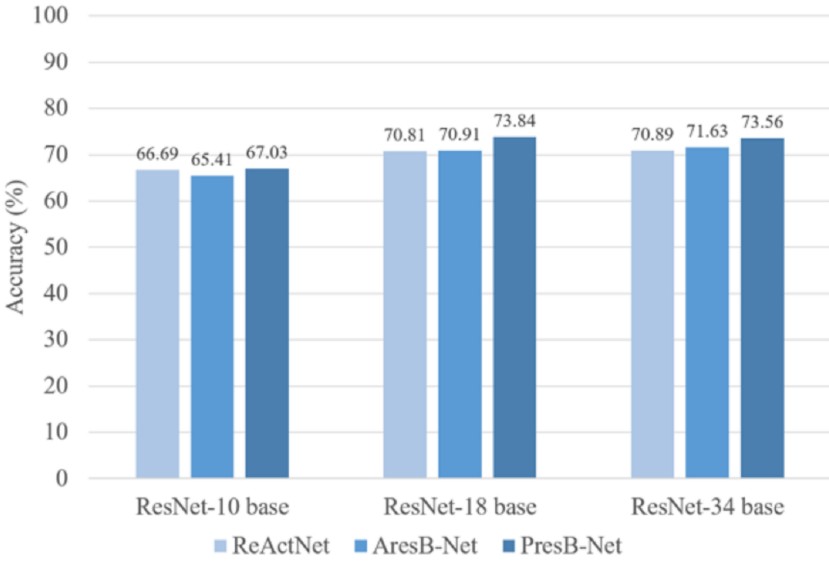

Figure 12 Comparison of accuracies (%) on CIFAR 100 dataset.

performance. Similarly, compared with ReActNet, the grouped shuffled convolution and new proposed normalization could help increase performance. In addition, PresB-Net shows better classification accuracy than several existing real-valued lightweight models such as ShuffleNet and MobileNet. The classification result of PresB-Net-18 is 11.3% and 14.2% higher than those of ShuffleNet (62.5% Top-1 accuracy) and MobileNet (59.634% Top-1 accuracy), respectively. Moreover, Top-1 accuracy of PresB-Net-18 is only 1.77% lower than 75.61% Top-1 accuracy of the real-valued ResNet-18. The summary of experimental results on CIFAR-100 dataset and comparisons between PresB-Net and other existing models are listed in Table 3 below. Accuracies of the proposed PresB-Net and other counterparts are illustrated in Fig. 12.

We additionally proceeded with an experiment on ImageNet dataset (*Russakovsky et al., 2015b*) which contains 1.2 million training and 50,000 validation color images classified into 1,000 categories. We resized images into 256 × 256 images and cropped the original image into 224 × 224 with a scale of 0.446 0.875. And we applied horizon flip to data argumentation. This same argumentation scheme is shown in *Kim (2021)* and

**Table 3  Comparison in terms of accuracies (%) on CIFAR-100 dataset.**

| Model accuracy | Model | Top-1 (%) | Top-5(%) | Top-1 gap (%) |
|---|---|---|---|---|
| Full-precision model | ShuffleNet | 62.51 | 82.60 | 11.33 |
| | MobileNet | 59.63 | 79.88 | 14.21 |
| | ResNet-18 | 75.61 | 93.05 | −1.77 |
| Binary-precision model | ReActNet-10 | 66.69 | 85.83 | 0.34 |
| | ReActNet-18 | 70.81 | 89.00 | 3.03 |
| | ReActNet-34 | 70.89 | 89.24 | 2.67 |
| | AresB-Net-10 | 65.41 | 88.22 | 1.63 |
| | AresB-Net-18 | 70.91 | 90.04 | 2.93 |
| | AresB-Net-34 | 71.63 | 88.85 | 1.93 |

**Table 4  Comparison in terms of accuracies (%) on ImageNet dataset.**

| Model accuracy | Model | Top-1 (%) | Top-5(%) | Top-1 gap (%) |
|---|---|---|---|---|
| Full-precision model | ResNet-18 | 69.3 | 89.2 | −6.3 |
| Binary-precision model | ReActNet-18 | 60.4 | 82.2 | 2.6 |
| | AresB-Net-18 | 54.8 | 78.2 | 8.2 |
| | PresB-Net-18 | 63.0 | 83.7 | – |

*He et al. (2016)*. The number of overall training epochs was set to 100 and as same as training strategy in CIFAR-100 dataset, and, 50 epochs proceeded with real-valued initial weights. We trained our proposed model PresB-Net-18 with ADAM optimizer(*Kingma & Ba, 2014*) with the learning rate of 0.1. When producing real-valued weights, not in binary operation, we used the weight decay of 1e−5. These training results of PresB-Net-18 and other counterparts are shown in Table 4. Our PresB-Net-18 achieves 63.0% Top-1 accuracy, which is 8.2% and 2.6% higher than 54.8% of AresB-Net and 60.4% of our ReActNet evaluation, respectively. Moreover, the Top-1 accuracy of PresB-Net-18 is 63.0% lower than 69.3% Top-1 accuracy of the real-valued ResNet-18 by 6.3%.

In these results, PresB-Net has outstanding accuracy to our evaluation of ReActNet and AresB-Net in the ImageNet dataset. Same as a reason in CIFAR-100 datasets results, proposed normalization method and new activation which is not applied to AresB-Net make PresB-Net show outstanding accuracy on various datasets.

The computation speed and memory usage of PreB-Net are analyzed as follows. Layer normalization does not consume additional memory space, so it only requires the computation overhead that calculates mean and variance. Since it has negligible overhead compared with the computational intensive convolution operation, we approximate the calculated FLOPs using the method described in *Kim (2021)*. Our PresB-Net model achieves dramatical speed up over real-valued ResNet-18 by $\frac{FLOPS(ResNet-18)}{FLOPS(PresB-Net-18)} \approx 36.8$ on CIFAR datasets. The difference in the number of channels of the first layer and the additional shortcut make computational speed slower than the AresB-Net-18 by $\frac{FLOPS(AresB-Net-18)}{FLOPS(PresB-Net-18)} \approx 0.9$. However, the proposed PresB-Net shows better classification results compared with those of AresB-Net. In terms of memory usage, PresB-Net-18

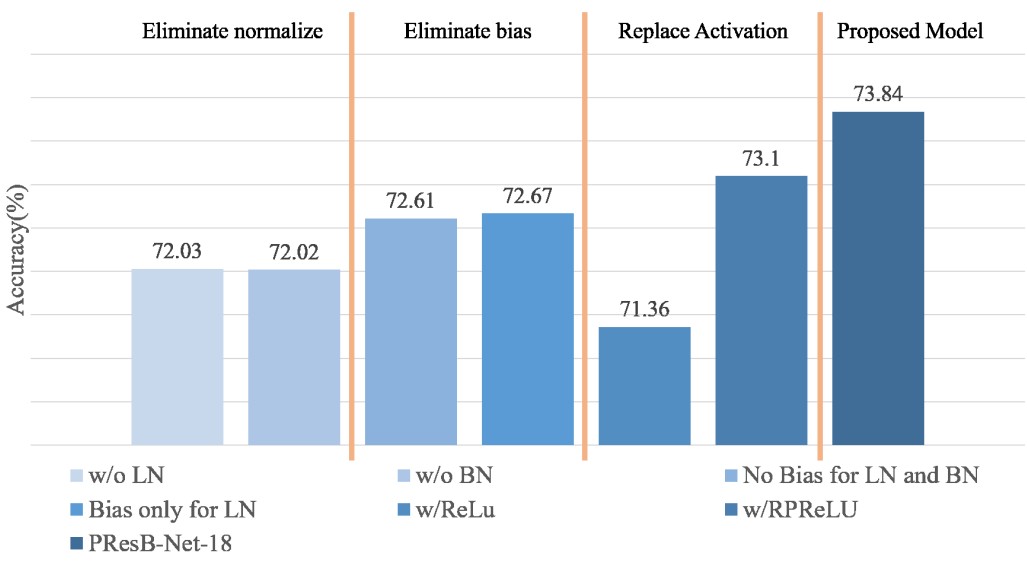

**Figure 13** **Summary of ablation studies.**

can reduce storage usage over real-valued ResNet-18 by $\frac{Storage(PresB-Net-18)}{Storage(ResNet-18)} \approx 0.038$ on CIFAR datasets. It is noted that PresB-Net-18 has storage usage similar to AresB-Net with remarkable increasing classification results.

## Ablation studies

Ablation studies are summarized in Fig. 13 and Table 5, containing the results of accuracy with the use of learnable biases and PReLU layers in Eq. (1). Terms LN and BN denote layer and batch normalizations, respectively. Term Bias means the learnable bias before normalizations. In the ablation study, PresB-Net-18 was evaluated, where several layers were modified to know their effectiveness. When learnable biases were not adopted before layer and batch normalizations(No Bias for LN and BN in Fig. 13), Top-1 accuracy was 72.61%, which is lower than the proposed model by 0.62%. When learnable biases were applied only before layer normalization(Bias only for LN in Fig. 13), Top-1 accuracy was 72.67%, which was 0.56% lower than that using learnable biases before both layer and batch normalizations. It was concluded that the additional learnable parameters of the biased PReLUs help increase Top-1 accuracies compared with the cases only using PReLUs.

Furthermore, Fig. 13 and Table 5 summarizes the effects of batch and layer normalizations in training stage. When layer normalization was not adopted (w/o LN in Fig. 13), Top-1 accuracy was 72.03%, showing 1.2% accuracy drop compared with the model using the layer normalization. The absence of layer normalization can bring imbalances between groups, which it was expected to make gradient convergence difficult. When batch normalization was removed, Top-1 accuracy was only 72.02%, showing 1.21% accuracy drop. The absence of batch normalization also made gradient convergence difficult due to its rough optimization field, (*Santurkar et al., 2018*) which make convergence difficult. As training does not work well even when learning parameters are applied, it can

**Table 5 Summary of inference accuracies(%) on varying structures in ablation studies.**

| Model | 1. Activation | 2. Normalization | 3. Activation | 4. Normalization | Top-1 |
|---|---|---|---|---|---|
| PresB-Net-18 | Biased PReLU | LN | Biased PReLU | BN | 73.84 |
| No Bias for LN and BN | – | LN | – | BN | 72.61 |
| Bias only for LN | Biased PReLU | LN | – | BN | 72.67 |
| w/o LN | Biased PReLU | – | Biased PReLU | BN | 72.03 |
| w/o BN | Biased PReLU | LN | Biased PReLU | – | 72.02 |
| w/RPReLU | RPReLU | LN | RPReLU | BN | 73.10 |
| w/ReLU | ReLU | LN | ReLU | BN | 71.36 |

be seen that it is also important to apply suitable normalization techniques like proposed normalization method. From these experimental results, we evaluated the cases when each activation or normalization was removed. In these experiments, the highest accuracy was achieved when the proposed PresB-Net-18 was adopted.

It is noted that RPReLU contains the first learnable bias, PReLU, and the second learnable bias layers in order. The biased PReLU did not use the second learnable bias, so that term $\zeta_i$ is removed from Eq. (1) of RPReLU. When applying RPReLU (*Liu et al., 2020*) instead of the biased PReLU, the final Top-1 accuracy was 73.10%, which was slightly lower than the case using the biased PReLU. When original ReLU is used instead of a biased PReLU, the final Top-1 accuracy is 71.36%, which is 2.48% lower than 73.86% from the proposed PresB-Net. This means that the biased PReLU could be usefully adopted for BNNs.

## CONCLUSION

This paper proposes a novel BNN structure called PresB-Net for achieving higher classification accuracy. The shuffled grouped convolution is applied to expand the channel with reduced computation resources. Our proposed normalization approach can solve the imbalance between groups in the grouped convolution. The new normalization technique adopts both layer and batch normalizations to normalize the entire channel and eliminate the imbalance between channel groups. Furthermore, we use the biased PReLU activation function with a learnable slope and binary activation with biases to improve performance. These normalizaiton, biased PReLU, and grouped shuffled convolution are used to construct basic blocks and expansion blocks. This paper described the proposed PreB-Net, which consists of stacked basic and expand blocks. Notably, the blocks have shortcuts for each binarized grouped convolution. Besides, there are shortcuts that skip two binarized grouped convolutions for each block. In our evaluation, PresB-Net-18 achieves 93.36% and 73.84% Top-1 final test accuracy on CIFAR-10 and CIFAR-100 datasets, respectively. When adopting same hyperparmeters and comparable structures, the proposed model can enhance Top-1 final accuracy 1.6%–2.4% over existing ReActNet and AresB-Net.

### Funding

The authors received no funding for this work.

### Competing Interests

The authors declare there are no competing interests.

### Author Contributions

- Jungwoo Shin conceived and designed the experiments, performed the experiments, analyzed the data, performed the computation work, prepared figures and/or tables, authored or reviewed drafts of the paper, and approved the final draft.
- HyunJin Kim analyzed the data, authored or reviewed drafts of the paper, and approved the final draft.

### Data Availability

The data is available at Github: https://github.com/SHINJUNGWOO/PresB-Net.

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
