# Peer review of "PresB-Net: parametric binarized neural network with learnable activations and shuffled grouped convolution"

_PeerJ Computer Science, doi:10.7717/peerj-cs.842_

## Round 0.1 · original submission · Major Revisions

The reviewers identified some major defects that need to be addressed in the revised version. I would like to recommend a major revision. Please prepare a detailed response letter and highlight the changes in the paper.

·

Basic reporting

pros:
The authors present a new binary network architecture, namely PresB-Net, which utilizes channel-shuffle and group binary convolution operations to increase the capacity of binary features without introducing extra computing costs. Besides, the proposed "Biased PReLU+LN+Biased PReLU+BN" module seems to solve the imbalance between different channel groups. Overall, the approach may shed some light on the design of compact binary networks and accurate BNN training.

I personally think the proposed method is simple yet effective. The technique part of this paper is easy to follow. The background and related works have been thoroughly discussed, such as group convolution, channel shuffle, and binary convolutions.

cons:
Since "group conv+channel shuffle" is a common setting in compact network design, I'm not sure whether PresB-Net is the first attempt to double the input channel number? The proposed "Basic block" seems to be a combination of ShuffleNet and DenseNet.

Experimental design

cons:
It would be better to fully evaluate the performance of PresB-Net, though it has achieved noticeable improvements over baseline results on CIFAR-100. Besides, I expect an ablation study shown in table form. The current manuscript makes it hard to understand the differences between the listed terms. Last but not least, the authors present a new architecture, however, I found no ablation study on the network design, e.g., why use "Biased PReLU+LN+Biased PReLU+BN" instead of "Biased PReLU+LN+BN"?

Validity of the findings

cons:
The authors focus on "fast output computation with low hardware costs" yet no real speedup over the FP32 counterpart has been reported in this paper. Note that when the mean and variance of LN are computed at inference time, the memory overhead and extra computing costs can be unacceptable in real-world applications.

Reviewer 2 ·

Basic reporting

This paper proposes PresB-Net, which applies shuffled grouped convolution to expand the channel with reduced computation resources. The proposed normalization approach can solve the imbalance between groups in the grouped convolution. Using the biased PreLU activation function with a learnable slope and binary activation with biases to improve performance. PresB-Net consists of stacked basic and expand blocks, the blocks have shortcuts for each binarized grouped convolution.
The proposed model can enhance Top-1 final accuracy 1.6%–2.4% over existing ReActNet and AresB-Net on CIFAR.

However, there are some issues in this paper which are listed as follows in the form of questions and suggestions.

1. Why is “performance degradation is the most critical problem in BNN models”.
2. The length of the paper is too small and the Section of experiments is rough. It is suggested to add some experimental contents, such as verifying your model on ImageNet, comparing with AresB-Net, ReActNet, MobileNet and ShuffleNet.
3. Whether using PreLU in ResNet can also improve performance? Whether the performance improvement of PresB-Net is related to PreLU?
4. “Therefore, the grouped binarized convolution results are normalized in both channel-wise and batch-wise manner. The proposed normalization approach allows the normalized convolutional results in both channel-wise and batch-wise manners, which can consider data correlations between groups.” Can the output of layer normalization and batch normalization be added directly? “consider data correlations between groups”, it needs to further explain how to consider relevance.
5. “solves the imbalance between groups arising from shuffled grouped convolution by considering the correlation of groups and allows the results of each group to follow a normal distribution, thereby increasing performance by balanced group and increasing the convergence.” How to solve imbalance between groups?
6. Whether can use basic and expand blocks to stack different network models?
7. Fig. 10 and 11 are too rough.
8. In the experiment, the reasons of the comparison results were not described.
9. The parameters of the model were not compared in the experiment.

Experimental design

no comment

Validity of the findings

no comment

Additional comments

no comment

---

## Round 0.2 · accepted · Accept

This paper can be accepted. Congratulations!

·

Basic reporting

I have carefully read the authors' feedback and comments from other reviewers. The authors have reflected the revision in the final version, hence I raise the overall score to an acceptance.

Experimental design

Whether the experiment results on CIFAR-100 are consistent with ImageNet? Note that the authors mainly conduct experiments on CIFAR-100 and it would be better to evaluate the correlation coefficient between them.

Validity of the findings

no comment

Reviewer 2 ·

Basic reporting

This paper proposes PresB-Net, which applies shuffled grouped convolution to expand the channel with reduced computation resources. The proposed normalization approach can solve the imbalance between groups in the grouped convolution. Using the biased PreLU activation function with a learnable slope and binary activation with biases to improve performance. PresB-Net consists of stacked basic and expand blocks, the blocks have shortcuts for each binarized grouped convolution. The proposed model can enhance Top-1 final accuracy 1.6%–2.4% over existing ReActNet and AresB-Net on CIFAR.

Experimental design

none

Validity of the findings

none

Additional comments

none